# L-Ornithine L-Aspartate Restores Mitochondrial Function and Modulates Intracellular Calcium Homeostasis in Parkinson’s Disease Models

**DOI:** 10.3390/cells11182909

**Published:** 2022-09-17

**Authors:** Maria Josè Sisalli, Salvatore Della Notte, Agnese Secondo, Carmelo Ventra, Lucio Annunziato, Antonella Scorziello

**Affiliations:** 1Division of Pharmacology, Department of Neuroscience, School of Medicine, University of Naples “Federico II”, 80131 Naples, Italy; 2ESSETI Farmaceutici, 80046 San Giorgio a Cremano, Italy; 3IRCCS Synlab SDN S.p.A, Via Gianturco 113, 80143 Naples, Italy

**Keywords:** mitochondria, calcium homeostasis, ornithine, NO, sodium-calcium-exchangers, Parkinson’s disease

## Abstract

The altered crosstalk between mitochondrial dysfunction, intracellular Ca^2+^ homeostasis, and oxidative stress has a central role in the dopaminergic neurodegeneration. In the present study, we investigated the hypothesis that pharmacological strategies able to improve mitochondrial functions might prevent neuronal dysfunction in in vitro models of Parkinson’s disease. To this aim, the attention was focused on the amino acid ornithine due to its ability to cross the blood–brain barrier, to selectively reach and penetrate the mitochondria through the ornithine transporter 1, and to control mitochondrial function. To pursue this issue, experiments were performed in human neuroblastoma cells SH-SY5Y treated with rotenone and 6-hydroxydopamine to investigate the pharmacological profile of the compound L-Ornithine-L-Aspartate (LOLA) as a new potential therapeutic strategy to prevent dopaminergic neurons’ death. In these models, confocal microscopy experiments with fluorescent dyes measuring mitochondrial calcium content, mitochondrial membrane potential, and mitochondrial ROS production, demonstrated that LOLA improved mitochondrial functions. Moreover, by increasing NCXs expression and activity, LOLA also reduced cytosolic [Ca^2+^] thanks to its ability to modulate NO production. Collectively, these results indicate that LOLA, by interfering with those mitochondrial mechanisms related to ROS and RNS production, promotes mitochondrial functional recovery, thus confirming the tight relationship existing between cytosolic ionic homeostasis and cellular metabolism depending on the type of insult applied.

## 1. Introduction

Parkinson’s disease (PD) is a chronic progressive neurodegenerative disease characterized by the demise of dopaminergic neurons in the *substantia nigra pars compacta* (SNc) and by the reduction in dopamine (DA) tone at the level of the *striatum* [1,2,3]. Specific neuropathological hallmarks of the disease are represented by the presence of intraneuronal protein aggregates called Lewy bodies and Lewy neurites [4], eosinophilic cellular inclusions comprising a dense core of filamentous material, which mainly consist of α-synuclein [5,6].

Although the aetiology of PD is still unknown, increasing evidence demonstrates that several factors, such as gender, mutation in specific genes, neuroinflammation, mitochondrial dysfunctions, oxidative stress, excitotoxicity, and dysfunction of the protein degradation system have all been shown to increase the risk of PD development [7,8,9,10]. More interestingly, the evidence described in the literature seems to converge on mitochondria as a primary target in the process of dopaminergic neuronal loss observed in PD [11,12,13]. Any alteration in the mitochondrial functionality seems to deeply affect the ability of cells to bear cellular stresses, thus appearing more susceptible to additional insults. Therefore, dysfunction in mitochondria results in a deficit supply of cellular energy and in a failure in maintaining ionic cellular homeostasis with a particular regard to calcium concentration. These events play a central role in the apoptotic and necrotic cell death pathway leading to neurodegeneration [14,15].

On these bases, a therapeutic approach aimed at reducing mitochondrial dysfunction might be considered useful to slow down dopaminergic neurons’ degeneration. However, the main limitation of this therapeutic strategy is to identify compounds able to selectively target mitochondria into their physiological intracellular environment. In this regard, ornithine, a non-essential amino acid, represents an interesting compound able to potentiate mitochondrial activity due to its ability to reach this cellular organelle so important for neuronal metabolism. Indeed, ornithine, once in the mitochondria, plays a key role in the urea cycle and creates an intermediate for arginine synthesis. Ornithine is produced by the enzymatic action of arginase on arginine, which results in the production of urea and generation of ornithine. Ornithine has been shown to reduce blood ammonia concentrations by increasing ammonia detoxification in the muscle and reducing the severity of hepatic encephalopathy in cirrhosis. Recent data have demonstrated the existence of the mitochondrial enzyme ornithine transcarbamylase (OTC) specifically in neurons positive for the nitric oxide synthesis enzyme (nNOS) [16], suggesting a potential role for ornithine in the modulation of brain functions. This hypothesis is supported by the observation that a defect in the ornithine translocase enzyme, responsible for the transport of ornithine at the mitochondrial level, determines a hyperornithinemia associated with hyperammonemia, homocitrullinemia, and neurological disorders accompanied, at the cellular level, by mitochondrial dysfunction and oxidative stress [17]. However, the molecular mechanisms through which ornithine at the mitochondrial level can perform actions to improve the redox and energy properties are not entirely understood.

On these premises, the present study has been undertaken to investigate the intracellular pathways affected by ornithine treatment in in vitro models of PD, with particular regards to cellular metabolism and mitochondrial function in order to identify new and selective therapeutic strategies to prevent the neuronal dysfunction occurring in PD, and consequently to slow down disease progression. This aspect is extremely relevant considering the key role played by mitochondria in regulating synaptic activity. Therefore, a therapeutic strategy able to promote mitochondrial function might be useful to counteract the early synaptic dysfunction and the functional and pathological changes occurring in the brain of PD-affected patients.

To this aim, in vitro experiments have been performed to evaluate the effect of L-Ornithine L-Aspartate (LOLA) on mitochondrial function in two cellular models of PD represented by SH-SY5Y cells treated with rotenone (ROT), an inhibitor of mitochondrial complex I, and 6-hydroxydopamine (6-OHDA), an inducer of oxidative stress. These experiments will be seminal for testing the effect of ornithine in an animal model recapitulating the prodromal phase of PD, and to demonstrate in vivo that improving mitochondria function might be useful to slow down PD progression.

## 2. Materials and Methods

### 2.1. Cell Culture

Human SH-SY5Y cells were cultured as monolayers in polystyrene dishes (100 mm, 60 mm, or 40 mm, according to the different experimental needs) in Dulbecco’s modified Eagle’s medium (DMEM) supplemented with 10% FBS, 1% penicillin (50 IU/mL), and streptomycin (50 μg/mL) in a humidified atmosphere at 37 °C with 5% CO_2_, and passed every 2–3 days in a humidified atmosphere of 5% CO_2_ and 95% air at 37 °C. The culture medium was replaced every 2 days. For confocal, microfluorimetry experiments and NO detection, SH-SY5Y cells were paced on glass coverslips pre-treated with poly-lysine 10 µM.

### 2.2. Reagents

L-Ornithine L-Aspartate (LOLA) (Esseti Farmaceutici, Italy) is a mixture of two endogenous amino acids: ornithine and aspartic acid. This combination makes ornithine more absorbable by the digestive system. This compound has the benefits of ornithine with the advantage that even aspartic acid, participating in the urea cycle, favors the elimination of ammonia. For these reasons, ornithine is commercialized in this form, because the two amino acids taken together are more effective in detoxification processes. Cells were exposed to 5 mM LOLA for 24 h or 48 h. The substance was dissolved in H_2_O.

Rotenone (ROT, Sigma-Aldrich, Italy) is an inhibitor of mitochondrial electron transport at nicotinamide adenine dinucleotide (NADH)-ubiquinone oxidoreductase. Rotenone acts as a neurotoxic agent, which can produce a Parkinson-like condition in both in vitro and in vivo models. Cells were exposed to 500 nM rotenone for 24 h. The substance was dissolved in DMSO (0.5 mg/mL).

Next, 6-OHDA (Sigma-Aldrich) is a neurotoxin, commonly used to induce PD both in in vitro and in vivo models. Indeed, 6-OHDA is highly oxidable and exerts cytotoxicity by generating reactive oxygen species, initiating cellular stress and cell death. Cells were exposed to 30 µM 6-OHDA for 48 h. The substance was dissolved in H_2_O.

### 2.3. Determination of Mitochondrial Oxidative Activity

Mitochondrial redox activity was assessed by measuring the level of mitochondrial dehydrogenase activity using the reduction of 3-(4,5-dimethylthiazol-2-yl)-2,5, diphenyltetrazolium bromide (MTT) as the substrate [18,19]. The assay was based on the ability of living mitochondria to convert dissolved MTT into insoluble formazan. Briefly, after treatments, the medium was removed, and the cells were incubated in 500 µM of MTT solution (0.5 mg/mL) for 1 h in a humidified 5% CO2 incubator at 37 °C. The incubation was then stopped by removing the medium and by adding 1 mL of DMSO to solubilize the formazan. The absorbance was detected at 540 nm. Data were expressed as the percentage of MTT reduction intensity compared to sham-treated cultures.

### 2.4. Confocal Microscopy and Mitochondrial Function

The mitochondrial membrane potential (ΔΨm) was assessed using the fluorescent dye tetramethylrhodamine ethyl ester (TMRE) in the “redistribution mode” [20]. Cells were loaded with TMRE (20 nM) for 30 min in a medium containing: 156 mM NaCl, 3 mM KCl, 2 mM MgSO_4_, 1.25 mM KH_2_PO_4_, 2 mM CaCl_2_, 10 mM glucose, and 10 mM HEPES. The pH was adjusted to 7.35 with NaOH. At the end of the incubation, cells were washed in the same medium containing TMRE (20 nM) and allowed to equilibrate. A decline in the mitochondria-localized intensity of fluorescence was indicative of mitochondrial membrane depolarization.

To assess the mitochondrial calcium concentrations [Ca^2+^]_m_, cells were loaded with X-Rhod-1 (0.2 µM) for 15 min in the above described medium. At the end of the incubation, cells were washed 3 times in the same medium. An increase in the mitochondria-localized intensity of fluorescence was indicative of mCa^2+^ overload [21].

Cytosolic calcium concentrations [Ca^2+^]_c_ were measured by using the fluorescent dye Fluo-3 acetoxymethyl ester (Fluo-3AM). The advantage of using Fluo-3AM was that this calcium indicator can be loaded into the cells together with the mitochondrial calcium indicator X-Rhod-1, or the mitochondrial membrane potential indicator TMRE, thus allowing a simultaneous comparison of calcium levels in the cytoplasm with the mitochondrial parameters. Cells were loaded with Fluo-3AM (5 nM) for 30 min at room temperature in the same medium described above. At the end of incubation, cells were washed 3 times in the same medium. An increase in the [Ca^2+^]_c_ intensity of fluorescence was indicative of cytosolic Ca^2+^ overload [10].

ROS levels were measured by using the fluorescent dye MitoSox (5 μM for 20 min) that is rapidly oxidized by superoxide. The oxidized product is highly fluorescent upon binding to nucleic acid. Cells were loaded with MitoSox (5 μM) for 20 min at room temperature in the same medium described above. At the end of incubation, cells were washed 3 times in the same medium. An increase in the MitoSox intensity of fluorescence was indicative of increased ROS production [22].

Confocal images were obtained using a Zeiss inverted 700 confocal laser scanning microscopy and a 63X objective. The illumination intensity of the 543 Xenon laser, used to excite X-Rhod-1, TMRE, and MitoSox; and the illumination intensity of the 488 Argon laser, used to excite Fluo-3AM fluorescence, were kept to a minimum of 0.5% of laser output to avoid phototoxicity. The quantification of fluorescence intensity was performed by using IMAGEJ software.

### 2.5. Western Blot Analysis

SH-SY5Y cells were lysed in a buffer containing 20 mM Tris-HCl (pH 7.5); 10 mM NaF; 150 mM NaCl; 1 mM phenylmethylsulphonyl fluoride (PMSF); 1% NONIDET P-40, 1%; 1 mM Na_3_VO_4_; 0.1% aprotinin; 0.7 mg/mL pepstatin; and 1 μg/mL leupeptin. Homogenates were centrifuged at 12.000 rpm for 30 min at 4 °C. The supernatant was collected and used for protein content quantification by the Bradford method. The total protein amount used for each sample was 50 μg. Proteins were separated on 10% sodium dodecyl sulphate polyacrylamide gels with 5% sodium dodecyl sulphate stacking gel (SDS-PAGE), and subsequently transferred to nitrocellulose membranes and incubated overnight at 4 °C in the blocking buffer containing 1:1000 antibody for ORNT1 (polyclonal rabbit antibody), 1:1000 antibody for NCX1 (polyclonal rabbit antibody), 1:1000 antibody for NCX3 (polyclonal rabbit antibody), and 1:1000 antibody for nNOS (polyclonal rabbit antibody). Next, all membranes were washed 3 times with a solution containing Tween 20 (0.1%), and subsequently incubated with the secondary antibodies for 1 h (1:2000) at room temperature. The immunoreactivity of the bands was visualized by enhanced chemiluminescence (ECL). The optical density of the bands was normalized with those of β-actin and measured by Image J.

### 2.6. [Ca^2+^]_i_ Measurement

SH-SY5Y cells, grown on glass coverslips, were loaded with 5 µM Fura-2 acetoxymethyl ester (Fura-2AM) for 1 h at room temperature in normal Krebs solution containing (in mM): 5.5 KCl, 160 NaCl, 1.2 MgCl_2_, 1.5 CaCl_2_, 10 glucose, and 10 Hepes–NaOH, pH 7.4. At the end of the Fura-2AM loading, the coverslips were placed into a perfusion chamber mounted onto the stage of an inverted Nikon Diaphot fluorescence microscope. A 100-Watt Xenon lamp, with a computer-operated filter wheel bearing two different interference filters (340 and 380 nm), illuminated the microscopic field with UV light every 3 sec, alternating the wavelengths at an interval of 500 ms. The light emitted by the Fura-2AM loaded cells was passed through a 400 nm dichroic mirror filtered at 510 nm and collected with an intensified camera. Images were digitized and analyzed with a Magiscan image processor driven by the AUTOLAB software (version 7.5.3, New York, NY, USA) [23]. NCX activity was evaluated as Ca^2+^ uptake through the reverse mode by switching the normal Krebs medium to the Na^+^- deficient NMDG+ medium (Na^+^- free) (in mM): 5.5 KCl, 147 N-methyl glucamine, 1.2 MgCl_2_, 1.5 CaCl_2_, 10 glucose, and 10 Hepes–NaOH (Ph 7.4).

### 2.7. Nitric Oxide Detection

Cells were loaded with 10 μM 4,5-diaminofluorescein-2-diacetate (DAF-2DA) in a humidified 5% CO_2_ atmosphere at 37 °C for 20 min in the normal Krebs’ solution (5.5 mM KCl, 160 mM NaCl, 1.2 mM MgCl2, 1.5 mM CaCl2, 10 mM glucose, and 10 mM HEPES-NaOH, pH 7.4) containing the drugs or their vehicles [24]. Thereafter, fluorescent cells were fixed with 4% (*w*/*v*) paraformaldehyde in phosphate-buffered saline for 5 min at 4 °C. This procedure allowed us to perform a subsequent densitometry analysis with the fluorescence microscope Nikon Eclipse E400 (Nikon, Torrance, CA, USA) set at an excitation/emission wavelength of 495/515 nm. Fluorescent images were then stored and analyzed with Pro-Plus software (Media Cybernetics, Silver Spring, MD). Data were calculated as the percentage of sample fluorescence compared with that of the controls.

Furthermore, time-lapse experiments were also carried out with the same digital imaging system used for [Ca^2+^]_i_ measurement. After loading with DAF 2-DA, cells were rapidly illuminated with 495 nm emitted from a Xenon lamp. The emitted light was passed through a 512 nm barrier filter. DAF-monitored NO● levels were expressed as arbitrary units of fluorescence.

### 2.8. Statistical Analysis

All data reported were generated from a minimum of 3 independent experimental sessions. Mitochondrial membrane potential, calcium, and free radical measurements were performed at least in 200 cells for each of the 3 independent experimental sessions. Specifically, 3–5 different fields were acquired for each experimental point. NCXs activity was measured by using Fura-2. The experiments were performed three times in at least 20 cells for each independent experiment. In all the figures, data were expressed as mean of the percentages ± S.E.M.

Statistical comparisons between the control and treated cells were performed using the one-way ANOVA test followed by the Newman–Keuls test. Statistical significance was accepted at the 95% confidence level. A *p*-value < 0.05 was considered statistically significant.

## 3. Results

### 3.1. LOLA Treatment Improves Mitochondrial Redox Activity in SH-SY5Y Cells Treated with ROT and 6-OHDA

To outline the pharmacological profile of LOLA, mitochondrial redox activity was evaluated in SH-SY5Y cells in basal conditions, and after the exposure to ROT (500 nM/24 h), or 6-OHDA (30 µM/48 h), two experimental conditions were largely validated to reproduce in vitro a mitochondrial dysfunction useful to investigate the contribution of mitochondria to the metabolic impairment involved in PD pathogenesis. As reported in Figure 1, long exposures to LOLA at different concentrations (500 nM–5 mM) for 24 h did not induce mitochondrial toxicity in SH-SY5Y cells until 1 mM, whereas, at 5 mM concentration, LOLA increased the redox activity of mitochondria, thus becoming a useful running concentration for all the experiments reported in this study (Figure 1 inset).

Indeed, as described in Figure 1A, the co-exposure of cells to LOLA 5 mM and 500 nM ROT for 24 h significantly reduced the detrimental effect of ROT on mitochondrial redox activity in SH-SY5Y cells (Figure 1A). Conversely, LOLA (5 mM) failed to revert the effect of 6-OHDA on mitochondrial redox activity in SH-SY5Y cells (Figure 1B). However, these effects were not related to a difference in the amount of ornithine entering into the cells in these two different experimental conditions, since the expression of the mitochondrial ornithine transporter 1 (ORNT1) did not change in cells exposed to 24 h ROT or 48 h 6-OHDA in the presence of LOLA (Figure 1C,D).

### 3.2. Mitochondrial Dysfunction Induced by ROT and 6-OHDA Exposure in SH-SY5Y Cells Is Counteracted by LOLA Treatment

In order to go deeper in the comprehension of the intracellular events involved in the effect of LOLA on mitochondria, confocal microscopy experiments using fluorescents probes such as TMRE, x-Rhod-1, and MITOSOX were performed to measure mitochondrial membrane potential, calcium content, and free radical production, respectively, in SH-SY5Y exposed to ROT or 6-OHDA in the presence and in the absence of LOLA. As reported in Figure 2, the treatment with LOLA was able to counteract ROT- and 6-OHDA- induced mitochondrial depolarization in SH-SY5Y cells by promoting a significant hyperpolarization of the mitochondrial membrane potential in both experimental conditions (Figure 2A,B).

This effect was accompanied by a reduction in LOLA-induced mitochondrial calcium content in SH-SY5Y cells exposed to ROT and to an increase in LOLA-induced mitochondrial calcium levels after the treatment with 6-OHDA (Figure 2C,D). Since mitochondrial calcium content is tightly regulated by cytosolic calcium concentration as well as by the mitochondrial membrane potential and ROS production [25,26], further experiments were performed in order to evaluate the effect of LOLA on cytosolic calcium concentrations and mitochondrial free radical production in SH-SY5Y cells exposed to ROT or 6-OHDA in the presence of LOLA. The results of these experiments demonstrated that LOLA significantly reduced cytosolic calcium levels in SH-SY5Y cells exposed to ROT, bringing them closer to those observed in the control untreated cells, whereas it did not modify intracellular calcium concentration in cells exposed to 6-OHDA (Figure 3A,B), thus suggesting that the LOLA effect on mitochondrial function might be related to its ability to interfere with the intracellular events affecting mitochondrial membrane potential. These data were supported by further experiments aimed to evaluate the effect of LOLA on mitochondrial free radical production. Indeed, LOLA treatment was able to slightly but significantly prevent the increase in mitochondrial ROS production induced by the exposure of SH-SY5Y cells to ROT (Figure 2E,F), whereas it was unable to decrease ROS levels in SH-SY5Y cells exposed to 6-OHDA. On the other hand, LOLA alone did not modify free radical production in basal conditions either after 24 h or after 48 h (Figure 2E,F). This finding, in line with data reported in the literature, supports the hypothesis that ROS production is strictly related to the metabolic activity of mitochondria that, in turn, depends on intracellular calcium concentration. Indeed, due to its ability to activate mitochondrial oxidative metabolism, cytosolic calcium promotes mitochondrial respiration [20,27,28,29,30]. Therefore, it is possible to speculate that 6-OHDA, by stimulating mitochondrial calcium accumulation in SH-SY5Y cells, promotes ROS production and consequently induces mitochondrial membrane depolarization, which represents the only mitochondrial parameter to be counteracted by LOLA treatment in cells exposed to 6-OHDA. On the contrary, ROT, by blocking mitochondrial complex I activity, primarily induces a massive mitochondrial membrane depolarization and free radical production; all effects that are counteracted by LOLA treatment in cells co-exposed to ROT.

### 3.3. LOLA Treatment Differently Regulates the Effects of ROT- and 6-OHDA on NCX1 and NCX3 Expression and Activity in SH-SY5Y Cells

To further understand the molecular mechanisms involved in the different effect of LOLA treatment on mitochondrial function in cells exposed to ROT or 6-OHDA, the attention was focused on the sodium calcium exchangers (NCXs) isoforms 1 and 3 (NCX1 and NCX3): two proteins responsible for the regulation of intracellular calcium homeostasis, since previously, data demonstrated their role in controlling calcium concentration in cytosol and in the mitochondria, respectively [23,31]. Therefore, Western blotting experiments were performed to evaluate the expression of the main neuronal isoforms NCX1 and NCX3 in SH-SY5Y cells treated with ROT or 6-OHDA in the presence of LOLA. As reported in Figure 3C,D, ROT induced an increase in NCX1 protein expression that was not affected by LOLA, whereas 6-OHDA did not affect NCX1 protein expression both when it was administered alone and when it was administered in the presence of LOLA.

On the contrary, both ROT and 6-OHDA alone stimulated NCX3 protein expression levels in SH-SY5Y treated cells; however, the co-treatment with LOLA counteracted the effect of 6-OHDA on NCX3 expression without affecting ROT-induced NCX3 increased expression (Figure 3E,F).

In order to study the functional consequences of the different effects of LOLA on NCX1 and NCX3 expression, additional experiments were performed to explore NCX activity in SH-SY5Y cells exposed to ROT or 6-OHDA in the presence of LOLA. These experiments were performed on single cells loaded with Fura-2AM using a microfluorimetric approach and a validated protocol to activate the reverse mode of operation of NCX by exposing the cells to a Na^+^ free medium [32]. The results of these experiments demonstrated that LOLA alone was able to potentiate the reverse mode of the operation of NCX as well as ROT and 6-OHDA, as indicated in Figure 3G,H. However, when the cells were co-exposed to LOLA, the activity of NCX was reduced in cells co-treated with ROT, whereas it was potentiated in cells co-treated with 6-OHDA, thus suggesting a potential different effect of LOLA in the regulation of calcium homeostasis in these two experimental PD models (Figure 3G,H). These data allow confirming the hypothesis that LOLA affected mitochondrial function with a mechanism depending on the type of toxin used to induce mitochondrial injury. Furthermore, they are in line with the effect of ornithine on ROS production as reported in Figure 2, and with data reported in the literature regarding the mechanism of action of ornithine on mitochondria, since this amino acid is able to decrease ROS levels and to increase nitric oxide (NO) production through the inhibition of mitochondrial arginase [33].

### 3.4. LOLA Treatment Reduces NO Production in SH-SY5Y Cells Exposed to ROT and 6-OHDA

Because it has also reported that both ROS and NO are able to modulate NCX activity [34], the hypothesis that the effect of LOLA on NCX activation in the experimental conditions described in the present study might be related to its ability not only to regulate the level of mitochondrial ROS but also of RNS, nNOS expression and NO production were explored in cells treated with ROT or 6-OHDA in the presence and in the absence of LOLA. In these experimental conditions, Western blot experiments were firstly performed to measure nNOS expression levels in SH-SY5Y cells. As reported in Figure 4A,B, neither ROT or 6-OHDA were able to improve nNOS protein expression both alone and in the presence of LOLA.

However, either ROT or 6-OHDA were able to increase NO production as demonstrated by the experiments performed in SH-SY5Y cells loaded with fluorescent dye DAF that selectively revealed the amount of NO produced by the single cell. The same entity of AU variation was obtained by detecting DAF-2 fluorescence on fixed cell preparations previously exposed to the same treatments (data not shown). These apparent discrepancies between nNOS expression and its activity in the two experimental conditions explored can be explained considering the existence of an mtNOS, identified as an inner membrane integral protein of 130 kDa and as the transcript of nNOS, splice variant α, myristoylated and phosphorylated [35,36,37]. At the mitochondrial level, nNOS functionally interacts with Complex-I (C-I), and its activity influences NO production as does the mitochondrial metabolism [38,39,40]. The finding that in the present study the effects of ROT and 6-OHDA were more pronounced in cells exposed to ROT than to 6-OHDA, further supports this hypothesis. Another possibility is that the huge NO production induced by ROT may cause a compensatory reduction of nNOS expression via such a post-translational mechanism.

Interestingly, the co-treatment with LOLA in ROT- or 6-OHDA-exposed cells was able to reduce NO production in both conditions as compared to the effect induced by the single toxins alone. However, the amount of NO produced in SH-SY5Y cells exposed to ROT in the presence of LOLA was still elevated compared to that observed in the control untreated or LOLA-exposed cells (Figure 4C). Conversely, in cells treated with LOLA and 6-OHDA, the amount of NO produced was similar to that observed in the control and in LOLA-treated cells (Figure 4D).

These results allowed confirming the hypothesis that LOLA might exert its effects on mitochondria by interfering with those mitochondrial mechanisms related to ROS and RNS production, thus favouring mitochondrial functional recovery. However, in cells treated with ROT, the increased ROS production was accompanied by an increase in intracellular calcium concentration, probably due to the stimulation of ROS-sensitive NCX1 activity in the reverse mode of operation. These effects, associated with the block of mitochondrial complex I caused by ROT, contributed to mitochondrial membrane depolarization and consequently to mitochondrial dysfunction. In these conditions, the treatment with LOLA, by reducing ROS and promoting NO production, was able to improve mitochondrial function by stimulating mitochondrial calcium efflux probably through NCX3 activation. Conversely, in 6-OHDA-treated SH-SY5Y cells, the stimulation of NCX1 activity, by promoting mitochondrial metabolic activation, induced a massive ROS production that was not counteracted by the LOLA treatment. However, the reduction of RNS production observed in SH-SY5Y cells co-treated with LOLA and 6-OHDA might represent the mechanism by which LOLA, without affecting mitochondrial calcium efflux mechanisms, might prevent mitochondria depolarization and, in turn, mitochondrial-induced 6-OHDA dysfunction (Figure 5).

In 6-OHDA-treated SH-SY5Y cells (lower left panel), the stimulation of NCX1 activity in the reverse mode of operation, by promoting mitochondrial metabolic activation, induced massive ROS and RNS production. The reduction of RNS production observed in SH-SY5Y cells co-treated with LOLA and 6-OHDA (lower right panel) represents the mechanism by which LOLA, without affecting mitochondrial calcium efflux mechanisms, prevents mitochondria depolarization and, in turn, mitochondrial-induced 6-OHDA dysfunction.

## 4. Discussion

The results of the present study demonstrate that LOLA treatment is able to prevent mitochondrial dysfunction induced by ROT and 6-OHDA exposure: two experimental conditions mimicking in vitro the pathogenic mechanisms leading to PD. Interestingly, these data also suggest that LOLA’s ability in protecting damaged mitochondria seems to be related to the severity of the injuring insult, since it preserves mitochondrial function with a mechanism that is different for the two experimental conditions explored. Indeed, although ROT- or 6-OHDA-induced mitochondrial membrane depolarization is counteracted by LOLA co-exposure, the LOLA effect on both mitochondrial calcium content and ROS production is different in SH-SY5Y cells treated with ROT or 6-OHDA, respectively. LOLA induces a reduction of mitochondrial calcium content in cells exposed to ROT and an increase in mitochondrial calcium levels in cells exposed to 6-OHDA. Moreover, LOLA can reduce ROS production in cells treated with ROT, whereas it is unable to decrease ROS levels in SH-SY5Y cells co-exposed to 6-OHDA. These findings, in line with data reported in the literature, support the hypothesis that ROS production is strictly related to the metabolic activity of mitochondria that, in turn, depends on intracellular calcium concentration. Indeed, an increase in cytosolic calcium concentration stimulates mitochondrial oxidative metabolism and promotes mitochondrial respiration, thus controlling cell survival [27,28,29,30]. However, if the mitochondrial oxidative metabolism overcomes the ability of the organelle to compensate free radical production as consequences of increased respiration, the mitochondrial membrane potential dramatically falls, and the cells die. This is what occurs in cells treated with ROT or 6-OHDA as reported in the present study. Interestingly, LOLA, due to the ability to differently affect intracellular calcium homeostasis and mitochondrial redox metabolism, represents a useful tool to improve mitochondrial performance in stress conditions. Indeed, the discrepancy in the effects of LOLA on ROT- or 6-OHDA-injured mitochondria might be explained just considering the different effectiveness of LOLA on cytosolic calcium concentrations in cells treated with the two toxins. Indeed, LOLA co-exposure significantly reduces cytosolic calcium levels in SH-SY5Y cells treated with ROT, bringing them closer to the levels observed in control untreated cells, whereas it does not affect intracellular calcium concentration in cells treated with 6-OHDA. Because previously, data demonstrated a tight relationship between calcium concentration in cytosol and in mitochondria, respectively [25,26], and considering that the sodium calcium isoforms NCX1 and NCX3 are among the cellular players involved in the regulation of this phenomenon in neurons [31,32,34], it is possible to speculate that LOLA may interfere with the molecular events controlling the expression and the activity of these two exchangers. This hypothesis is further supported by the finding that NCX3, apart from its expression at the plasma membrane level, is also detectable on the outer mitochondrial level, where it plays a role in the regulation of mitochondrial calcium efflux in physiological and in pathological conditions such as ischemia and PD [10,31,41,42], and that NCX1 activation represents a neuroprotective mechanism in ischemic conditions [43,44]. Moreover, NCX1 and NCX3 have been considered as causal factors to neurodegeneration and neuroinflammation in an in vivo model of PD [10,45]. The results obtained in the present study demonstrate that an increase in NCX1 and NCX3 expression is detectable in cells treated with ROT, whereas in cells exposed to 6-OHDA, a rise in NCX3 protein expression occurs without any change in NCX1 expression. Moreover, both ROT and 6-OHDA are able to improve the reverse mode of operation of sodium calcium exchanger, thus contributing to the increase in cytosolic calcium concentration in both the experimental conditions investigated. Interestingly, the co-exposure of cells to LOLA can prevent the effects of ROT on NCX3 expression without affecting 6-OHDA-induced effects on NCX1 and NCX3 expression. From a functional point of view, it is important to underline that the experiments performed in this study demonstrate that LOLA alone is able to potentiate the reverse mode of operation of the exchanger without altering mitochondrial oxidative metabolism. Interestingly, when the cells are co-exposed to LOLA and ROT or 6-OHDA, the activity of the exchanger is reduced in cells treated with ROT, whereas it is potentiated in cells treated with 6-OHDA. Because ROS and NO can modulate NCX activity [34,44], these effects may be related to the ability of LOLA to regulate the level of mitochondrial ROS and RNS, thus modulating mitochondrial function (Figure 5). This hypothesis is supported by the finding that in cells treated with ROT, the increased ROS production is accompanied by an increase in intracellular calcium concentration, probably due either to a direct inhibition of Complex I with a consequent alteration of mitochondrial membrane permeability, or to the stimulation of ROS-sensitive NCX1 activity in the reverse mode of operation which, in turn, contributes to further promote mitochondrial respiration. These effects, together, contribute to a severe mitochondrial membrane depolarization and consequently cause mitochondrial dysfunction. In these conditions, the treatment with LOLA, by reducing ROS and promoting NO production due to an amelioration of mitochondrial metabolism [36,37,38], can improve mitochondrial function stimulating mitochondrial calcium efflux through NCX3 activation. Conversely, in 6-OHDA-treated SH-SY5Y cells, the stimulation of NCX1 activity, by promoting mitochondrial metabolic activation, induces a massive ROS production that is not counteracted by LOLA treatment. However, the reduction of RNS production observed in SH-SY5Y cells co-treated with LOLA and 6-OHDA might represent the mechanism by which LOLA, without affecting mitochondrial calcium efflux mechanisms, might prevent mitochondria depolarization through the inhibition of the NCX1 reverse mode of operation, thus reducing intracellular calcium concentration and, in turn, mitochondrial dysfunction occurring in cells exposed to 6-OHDA treatment alone. Therefore, depending on the mechanism by which each toxin induces mitochondrial functional impairment, LOLA may contribute to the recovery of mitochondrial functional properties by affecting two different intracellular pathways: one direct on the mitochondria, finalized to counteract a ROT-induced mitochondrial calcium increase; and one direct to the plasmamembrane, aimed to reduce 6-OHDA-induced cytosolic calcium content. These findings, also supported by the effects of LOLA on RNS and ROS production, are in line with the hypothesis that these radical species are able to stimulate NCXs activity, thus playing a role in the regulation of intracellular calcium homeostasis [34,42,44]. This hypothesis, although it needs to be further confirmed, seems to suggest a novel emerging mechanism of action for LOLA that is not only directly related to its ability to improve mitochondrial metabolism, since it selectively enters into mitochondrial thanks to the ORNT1 transporter on the mitochondrial membrane, but is also dependent on the ability of LOLA to activate NO production, as already described in some brain nitrergic neurons [16]. Moreover, the results of the present study confirm the tight relationship existing between cytosolic ionic homeostasis and cellular metabolism and underline the importance to finely regulate calcium fluxes among different cellular compartments such as mitochondria, cytosol, and plasma membrane as mechanisms claimed to explain cellular vulnerability to stress conditions. Collectively, the results of the present study, allow us to conclude that ornithine might represent a new pharmacological strategy able to improve mitochondrial function and to prevent dopaminergic neuronal dysfunction. This mechanism may be useful in those pathological conditions in which mitochondrial dysfunction represents a pathogenic factor in disease development, such as that which occurs in PD.

## Figures and Tables

**Figure 1 cells-11-02909-f001:**
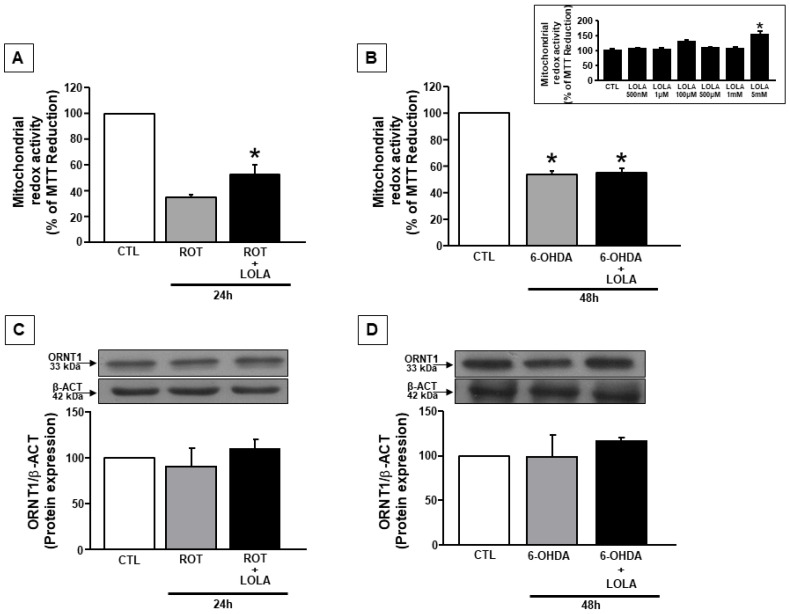
Mitochondrial redox activity in SH-SY5Y cells exposed to ROT and 6-OHDA and treated with LOLA: (**A**) Mitochondrial redox activity in cells treated with ROT (500 nM/24 h) and ROT+LOLA (5 mM/24 h). * *p* < 0.05 vs. ROT. (**B**) Mitochondrial redox activity in cells treated with 6-OHDA (30 µM/48 h) and 6-OHDA+LOLA; * *p* < 0.05 vs. CTL. (Inset) Dose-response of LOLA treatment for 24 h on mitochondrial redox activity in basal conditions. * *p* < 0.05 vs. CTL. (**C**) Expression of the ornithine transporter 1 (ORNT1) after exposure to ROT. (**D**) Expression of the ornithine transporter 1 (ORNT1) after exposure to 6-OHDA. The values for each column represent the mean of the percentage ± SEM.

**Figure 2 cells-11-02909-f002:**
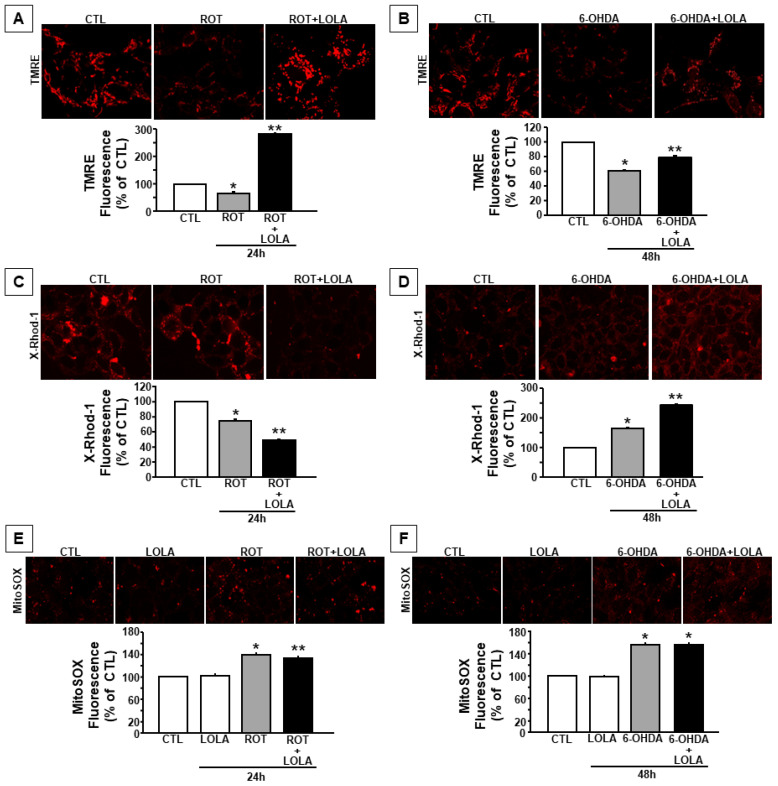
Mitochondrial functional parameters in SH-SY5Y cells exposed to ROT and 6-OHDA in the presence or in the absence of LOLA: (**A**) Quantification of mitochondrial membrane potential in cells treated with ROT (500 nM/24 h) and ROT+LOLA (5 mM/24 h). * *p* < 0.05 vs. CTL, ** *p* < 0.05 vs. ROT. (**B**) Quantification of mitochondrial membrane potential in cells treated with 6-OHDA (30 µM/48 h) and 6-OHDA+LOLA. * *p* < 0.05 vs. CTL, ** *p* < 0.05 vs. 6-OHDA. (**C**) Quantification of [Ca^2+^]_m_ in cells treated with ROT and ROT+LOLA. * *p* < 0.05 vs. CTL, ** *p* < 0.05 vs. ROT. (**D**) Quantification of [Ca^2+^]_m_ in cells treated with 6-OHDA and 6-OHDA+LOLA. * *p* < 0.05 vs. CTL, ** *p* < 0.05 vs. 6-OHDA. (**E**) Quantification of ROS production in cells treated with LOLA, ROT, and ROT+LOLA. * *p* < 0.05 vs. CTL, LOLA; ** *p* < 0.05 vs. ROT. (**F**) Quantification of ROS production in cells treated with LOLA, 6-OHDA, and 6-OHDA+LOLA. * *p* < 0.05 vs. CTL, LOLA. Mitochondrial functional parameters were measured at least in 200 cells for each of the 3 independent experimental sessions. Data were expressed as percentage ± S.E.M. Statistical comparisons between control and treated cells were performed using the one-way ANOVA test followed by Newman–Keuls test. The values for each column represent the mean of the percentages ± SEM.

**Figure 3 cells-11-02909-f003:**
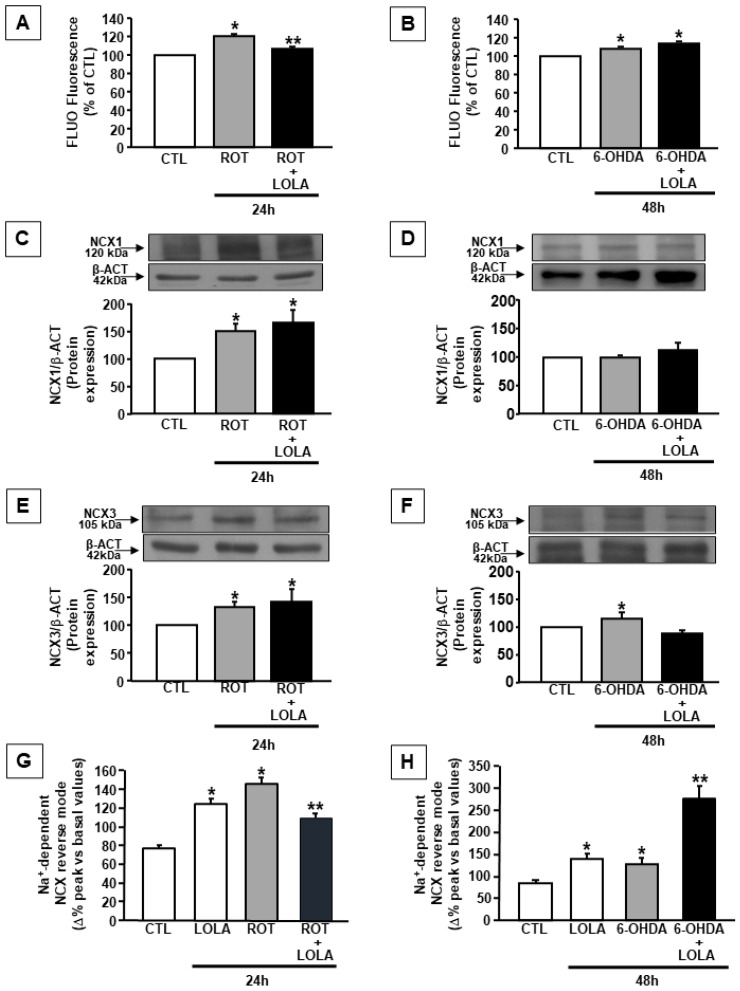
[Ca^2+^]_c_ and NCXs expression and activity in SH-SY5Y cells exposed to ROT and 6-OHDA in the presence and in the absence of LOLA: (**A**) Quantification of [Ca^2+^]_c_ in cells treated with ROT (500 nM/24 h) and ROT+LOLA (5 mM/24 h). * *p* < 0.05 vs. CTL; ** *p* < 0.05 vs. ROT. (**B**) Quantification of [Ca^2+^]_c_ in cells treated with 6-OHDA (30 µM/48 h) and 6-OHDA+LOLA. * *p* < 0.05 vs. CTL. (**C**) NCX1 expression in SH-SY5Y cells treated with ROT and ROT+LOLA and (**D**) with 6-OHDA and 6-OHDA+LOLA; * *p* < 0.05 vs. CTL. (**E**) NCX3 expression in cells treated with ROT and ROT+LOLA and (**F**) with 6-OHDA and 6-OHDA+LOLA. * *p* < 0.05 vs. CTL. (**G**) NCXs activity in cells treated with LOLA, ROT, and ROT+LOLA and (**H**) with LOLA, 6-OHDA, and 6-OHDA+LOLA, measured in at least 20 cells for each independent experiments. All experiments were performed in three different experimental sections. Each column represents the mean of the percentage ± SEM. * *p* < 0.05 vs. CTL, ** *p* < 0.05 vs. ROT or 6-OHDA.

**Figure 4 cells-11-02909-f004:**
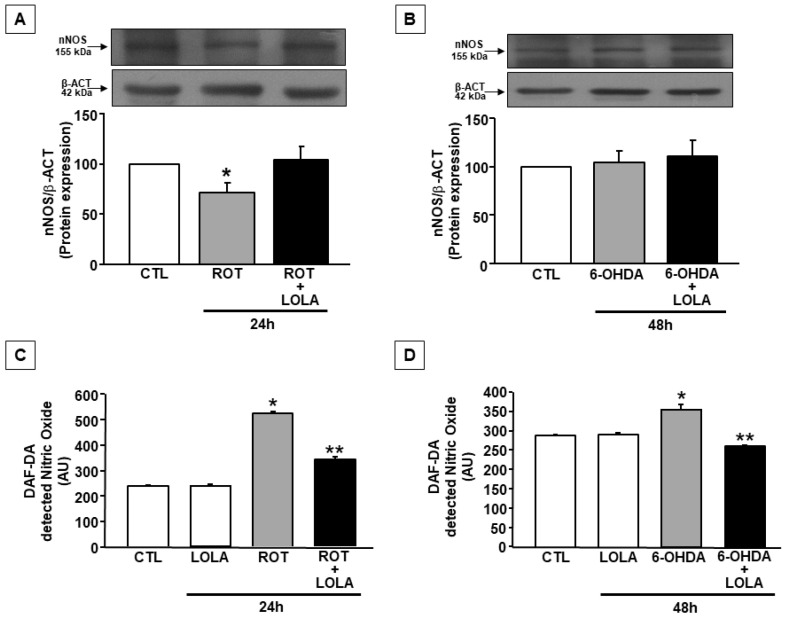
nNOS expression and NO production in SH-SY5Y cells exposed to ROT and 6-OHDA and treated with LOLA: (**A**) nNOS expression in cells treated with ROT (500 nM/24 h) and ROT +LOLA (5 mM/24 h) and (**B**) with 6-OHDA (30 µM/48 h) and 6-OHDA+LOLA; * *p* < 0.05 vs. CTL. (**C**) NO production in cells treated with LOLA, ROT, and ROT+LOLA; * *p* < 0.05 vs. CTL; ** *p* < 0.05 vs. ROT and CTL. (**D**) NO production in cells treated with LOLA, 6-OHDA, and 6-OHDA +LOLA; * *p* < 0.05 vs. CTL; ** *p* < 0.05 vs. 6-OHDA. The values are expressed as arbitrary units measured on 4 different experiments. In panels (**A**,**B**), each column represents the mean of percentage ± SEM; in panels (**C**,**D**), each column represents the mean ± SEM.

**Figure 5 cells-11-02909-f005:**
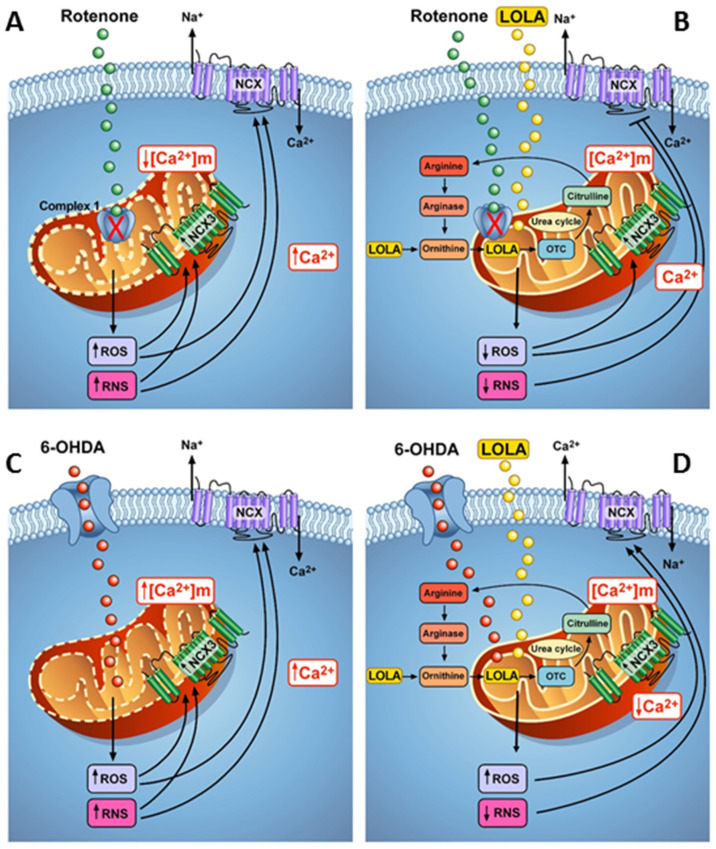
Schematic model of LOLA effects on mitochondrial activity and [Ca^2+^]_i_ in SH-SY5Y cells exposed to ROT and 6-OHDA. In cells treated with ROT (**A**–**C**), the increased ROS production causes the increase in intracellular calcium concentration probably due to the stimulation of ROS-sensitive NCX1 activity in the reverse mode of operation. These effects, associated with the block of mitochondrial complex I contributed to mitochondrial membrane depolarization and consequently mitochondrial dysfunction. In these conditions, the treatment with LOLA (**B**–**D**), by reducing ROS and NO production, improves mitochondrial function promoting mitochondrial calcium efflux through mitochondrial NCX3.

## Data Availability

Not applicable.

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
