# Peer review of "L-Ornithine L-Aspartate Restores Mitochondrial Function and Modulates Intracellular Calcium Homeostasis in Parkinson’s Disease Models"

_cells, 2022, doi:10.3390/cells11182909_

Round 1

Reviewer 1 Report

The authors present a pre-clinical model of Parkinsons Disease to understand the effect of the amino acid combination of l-ornithine and l-asparate (LOLA). The authors use SH-SY5Y cells treated with two chemicals that induce Parkinson like phenotypes to understand if LOLA could be used to rectifiy issues with mitochondria, calcium and nitric oxide signalling. 

Overall, the article is clearly written and follows a logical progression of results, but the comparison between the two different chemicals to induce Parkinson like phenotypes confuses the explanations being provided. The effects of LOLA on the phenotypes induced by the two chemicals are quite different. A summary figure to explain the molecular mechanisms of LOLA on the two types of chemicals effects would be extremely informative to help clarify this for the reader. This is especially true for the explanation of Figure 2 and the text found from lines 247 to 263. 

More specific comments are listed now:

1 - In the methods section L-ornithine L-aspartate is referred to as a "peptide". This implies the amino acids are covalently linked. LOLA is often sold as a combination of l-ornithine and l-aspartate as two separate salts combined. Can the authors clarify what form the LOLA is used?

2 - A number of figures have images that are quantified as the data used for histrograms (Figure 2). The authors need to provide a more thorough description of how the fluorescence images were analysed and quantified? How many images were selected and quantified? Were these biological and technical replicates? If so, what number of replicates were used? Were multiple images from the same sample analysed or was it just the one image that was presented? 

3 - In Figure 1 the y-axis  for panel A and B is written as "cell viability". As this is an MTT assay, the viability of cells may not be the only contributing factor to the reading. The y-axis could be simply labelled as % MTT reduction to avoid this overlap. If the authors want to determine viability, another approach should be used. 

4 - In Figure 4 there is a decrease in nNOS (panel A) after rotenone treatment, which is accompanied by an increase in NO (panel C), while in panel B there is no change in nNOS, but and increase in NO caised by 6-OHDA. Can the authors provide a deeper overview of how these two assays can be reconciled. What else could be altered to provide these seemingly inconsistent pieces of data. 

Author Response

Referee #1

  1. Point 1.: Overall, the article is clearly written and follows a logical progression of results, but the comparison between the two different chemicals to induce Parkinson like phenotypes confuses the explanations being provided. The effects of LOLA on the phenotypes induced by the two chemicals are quite different. A summary figure to explain the molecular mechanisms of LOLA on the two types of chemicals effects would be extremely informative to help clarify this for the reader. This is especially true for the explanation of Figure 2 and the text found from lines 247 to 263. 

Answer to point 1: As requested by the reviewer an additional figure to explain the proposed molecular mechanisms of LOLA on mitochondrial activity and calcium homeostasis in the two experimental models investigated has been included in the revised version of the manuscript. Please, see Fig 5 in the revised version. Furthermore, an explanation has been included to clarify the description of the results reported in the Figure 2. (Page 7, Lines 245-253 and Page 9, Lines 255-265)

  1. Point 2: In the methods section L-ornithine L-aspartate is referred to as a "peptide". This implies the amino acids are covalently linked. LOLA is often sold as a combination of l-ornithine and l-aspartate as two separate salts combined. Can the authors clarify what form the LOLA is used?

Answer to point 2:  We apologise for the mistake in referring to LOLA as a peptide rather than a combination of two amino acids. In the Material and Methods’ section of the revised version of the manuscript the form of LOLA used has been specified. (Page 3, lines 99-101)

  1. Point 3. A number of figures have images that are quantified as the data used for histograms (Figure 2). The authors need to provide a more thorough description of how the fluorescence images were analysed and quantified? How many images were selected and quantified? Were these biological and technical replicates? If so, what number of replicates were used? Were multiple images from the same sample analysed or was it just the one image that was presented? 

Answer to point 3: We apologise for omitting the description of quantification of the results reported in the Figure 2 as far as concern the fluorescence image and the statistical analysis. The omitted information are now indicated in the revised version of the manuscript in the Material and Methods’ section (Page 5; lines 206-215) and in the Figure 2 and in the other figures of the manuscript.

  1. Point 4. In Figure 1 the y-axis  for panel A and B is written as "cell viability". As this is an MTT assay, the viability of cells may not be the only contributing factor to the reading. The y-axis could be simply labelled as % MTT reduction to avoid this overlap. If the authors want to determine viability, another approach should be used. 

Answer to point 4.: we apologise for the mistake reported in the panels A and B of figure 1 considering MTT assay a method to evaluate cell viability. The aim of the experiments was to demonstrate the effects of the treatments with toxins and LOLA on mitochondrial function. Therefore, in the revised version of the manuscript cell viability was changed accordingly to the referee’s suggestion as Mitochondrial redox activity % of MTT reduction, both in the figure 1 and throughout the text.

  1. Point 5. In Figure 4 there is a decrease in nNOS (panel A) after rotenone treatment, which is accompanied by an increase in NO (panel C), while in panel B there is no change in nNOS, but an increase in NO raised by 6-OHDA. Can the authors provide a deeper overview of how these two assays can be reconciled? What else could be altered to provide these seemingly inconsistent pieces of data? 

Answer to point 5.: As requested by the referee, an explanation for the discrepancy in the results reported in Figure 4 (A and B v.s C and D) has been provided in the results‘section and in the discussion of the revised version of the manuscript. (Page 14; lines 365-375; Page 17; lines 484-486)

Reviewer 2 Report

The labels on the bar-chart are not straight forward. It is hard to read and find the annotation in the figure legend to figure out the significant stars’ meaning. Please label the stars’ meaning by labeling which 2 columns on the bar-chart are represented by the star.

The statistics are not fully done on all the tests. Please compare the results of all 3 groups with one-way ANOVA followed with the test between each of 2 columns.

The result part is not well written. It is hard to understand what the experiments want to show. Like TMRE and X-Rhod-1, they are only mentioned in the method part. It is very easy to briefly describe the meaning of experiments in the result.

In figure 1 inset, the MTT assay was conducted to reflect the cell viability. Please explain why MTT assay represents cell viability. Please explain why 24 hours was chosen for the ROT and ROT/LOLA test but 48 hours for the 6-OHDA and 6-OHDA/LOLA test. Please show the criteria of the experiments. Does 48 hours give better signal or is 48 hours more reliable?

The images in Figure 2 are too dim. It is very hard to tell any difference.

In figure 3E, column ROT is higher than column ROT+LOLA, however the quantification showed that ROT signal is lower than ROT+LOLA signal.  Figure 3C and 3D have the similar problem. If the western blot and the quantification showed the different results, which one reflects the real trend? 

Author Response

Referee #2

  1. Point 1. The labels on the bar-chart are not straight forward. It is hard to read and find the annotation in the figure legend to figure out the significant stars’ meaning. Please label the stars’ meaning by labeling which 2 columns on the bar chart are represented by the star.

Answer to point 1.: As requested by the referee the figures have been enlarged and the bar-chart modified accordingly. Moreover, the legends to the figure and the stars’ meaning related to the columns have been labelled in the revised version of the manuscript. See figures 1-4

  1. Point 2. The statistics are not fully done on all the tests. Please compare the results of all 3 groups with one-way ANOVA followed with the test between each of 2 columns.

Answer to point 2.: We apologise for omitting the statistics on all the tests. In the revised version of the manuscript a paragraph, including the statistical analysis and the tests used, has been added in the Method’s section. (Page 5; lines 206-215)

  1. Point 3. The result part is not well written. It is hard to understand what the experiments want to show. Like TMRE and X-Rhod-1, they are only mentioned in the method part. It is very easy to briefly describe the meaning of experiments in the result.

Answer to point 3.: As requested by the referee the description of the results has been improved with particular regard to the description and the meaning of the experiments performed with TMRE and X-Rhod-1. See results’ section of the revised manuscript (pages 7-9; lines 243-294)

  1. Point 4. In figure 1 inset, the MTT assay was conducted to reflect the cell viability. Please explain why MTT assay represents cell viability. Please explain why 24 hours was chosen for the ROT and ROT/LOLA test but 48 hours for the 6-OHDA and 6-OHDA/LOLA test. Please show the criteria of the experiments. Does 48 hours give better signal or is 48 hours more reliable ?

Answer to point 4.: We apologise for considering MTT assay a test used to measure cells viability rather than mitochondrial oxidative capacity. Due to the ability of the test to evaluate the activity of mitochondria that represents the powerhouse of the cells, an impairment of mitochondrial redox activity can indirectly be correlated with a reduction in cells survival. Therefore, we changed “cell viability” with “mitochondrial redox activity” in the figure 1 and throughout the text. Regarding the choice to perform the experiments with the toxins at two different time points, the explanation is related to the different strength of the two toxins in inducing mitochondrial damage, being ROT more potent than 6-OHDA. Specifically the dose response effect of 6-OHDA (10, 30 and 50 mM) did not induce any significant effect on mitochondrial redox activity after 24 hr of treatment, whereas after 72 hrs the percentage of mitochondrial redox capacity was greatly compromised. Similarly, the dose response effect of ROT : 500nM-1mM for 24 and 48 hr caused a massive reduction of mitochondrial redox activity. Therefore, the choice of concentrations and exposure times of cells to the toxins was based on the ability of each of them to induce a 50% reduction in mitochondrial redox capacity as it occurs with ROT 500nM 24hr and 6-OHDA 30mM 48 hr

  1. Point 5. The images in Figure 2 are too dim. It is very hard to tell any difference.

            Answer to point 5.: We apologise for the quality of the images reported in figure 2 that have been enlarged and substituted, when possible, with others brighter. See Fig 2 panels E and F, in the revised version of the manuscript.

  1. Point 6. In figure 3E, column ROT is higher than column ROT+LOLA, however the quantification showed that ROT signal is lower than ROT+LOLA signal.  Figure 3C and 3D have the similar problem. If the western blot and the quantification showed the different results, which one reflects the real trend? 

          Answer to point 6.: We apologise for the quality of the western blot  reported in figure 3 E that has been changed with another one more representative in the revised version of the manuscript.

Reviewer 3 Report

The manuscript by Sisalli et al. wishes to investigate the putative cytoprotective properties of L-ornithine-L-aspartate (LOLA) in a cell culture model of Parkinson’s disease. Although the hypothesis is of interest, this Reviewer finds a number of shortcomings preventing the evaluation of the merits and the appreciation of the findings.

Major concerns:

#1. Although the SH_SY5Y cells were exposed to various concentrations of LOLA to determine the effect on cell viability – in subsequent experiments only the highest – quite supramaximal 5 millimolar! concentration was used. The inset of Figure 1 clearly shows that this dose increased „viability” ~ 50% that can mean only that the biochemical reaction used for the viability test (MTT) reaction was directly affected by this high concentration of LOLA. In this case, alternative viability tests are warranted to determine actually the detrimental effects of the drugs especially rotenone and 6OHDA and potential protective effect of LOLA.

#2. In all experiments shown in Figs 1-3, LOLA was only co-applied with either rotenone or 6OHDA, although it is quite likely that 5mM LOLA would have had a direct effect on these measures (see above) . As these controls (treated with LOLA only for 24-48 hours) are missing throughout, the true mechanism of LOLA action cannot be read from the findings.

#3. Dose-response effect of LOLA are missing throughout the results. What would be the in vivo dose of LOLA to reach a 5mM concentration in the brain EC fluid (as the Introduction states that these experiments are seminal for future in vivo studies)?

#4. The number of independent experiments, the number of dishes/wells/cells etc in the performed experiments are not disclosed. As the bar of the controls is without a whisker in most graphs (no variability), it suggests n=1 at least for the controls.

#5. The statistics are not described at all, rendering the evaluation of the findings also impossible.

These shortcomings must be addressed before any further evaluation is possible.

Author Response

Referee # 3

  1. Point 1. Although the SH_SY5Y cells were exposed to various concentrations of LOLA to determine the effect on cell viability – in subsequent experiments only the highest – quite supramaximal 5 millimolar! Concentration was used. The inset of Figure 1 clearly shows that this dose increased „viability” ~ 50% that can mean only that the biochemical reaction used for the viability test (MTT) reaction was directly affected by this high concentration of LOLA. In this case, alternative viability tests are warranted to determine actually the detrimental effects of the drugs especially rotenone and 6OHDA and potential protective effect of LOLA.

Answer to point 1.: We understand the referee concern related to the effects of LOLA on mitochondrial activity in basal conditions. However, considering that ornithine enters into mitochondria and takes part to the urea cycle with consequent production of intermediate that can affect cellular oxidative metabolism, it is expectable that the effect observed with the higher ornithine concentration might be correlated to the improvement of mitochondrial redox activity as confirmed by 50% increase of MTT assay. Indeed, once in the mitochondria ornithine is transformed in citrulline by the enzyme ornithine trancarbomylase (OTC), which is able to stimulate complex-I activity. (Wu et al., European J of Physiology, 472:1743-1755, 2020).

  1. Point 2. In all experiments shown in Figs 1-3, LOLA was only co-applied with either rotenone or 6OHDA, although it is quite likely that 5mM LOLA would have had a direct effect on these measures (see above) . As these controls (treated with LOLA only for 24-48 hours) are missing throughout, the true mechanism of LOLA action cannot be read from the findings.

 Answer to point 2.:We apologize for the absence of LOLA control in Figures 1-3. In the revised version of the manuscript we added a column corresponding to LOLA effect (5mM) in basal conditions on mitochondrial parameters (Figure 2), cytosolic calcium concentration (Figure 3) and NO production (Figure 4) both at 24 and 48 hr depending on the respective toxin utilized in the experiments.

  1. Point 3. Dose-response effect of LOLA are missing throughout the results. What would be the in vivo dose of LOLA to reach a 5mM concentration in the brain EC fluid (as the Introduction states that these experiments are seminal for future in vivo studies)?

 Answer to point 3.:Regarding the referee concern in terms of correlation between the in vivo and in vitro dose of LOLA, it can be clarified using the classical In Vitro In Vivo Correlation (IVIVC) analysis. In particular, according to the Biopharmaceutics Classification System, which takes in consideration the chemical features of the compounds in terms of solubility and permeability a good IVIVC correlation is expected for LOLA, Indeed, due to its high solubility  and high permeability LOLA can be included among the class I compounds (Emami J, J Pharm Pharmaceut Sci 9 (2): 169-189,  2006; see also "Guidance Document on Using In Vitro Data to Estimate In Vivo Starting Doses for Acute Toxicity”). In this case, when it is possible to apply IVIVC analysis, to obtain an in vivo dose comparable to that effective in vitro, it is necessary to increase the in vitro range of 1 log differences. Therefore, the concentration of LOLA usable for in vivo study might be of gr/kilo, a dosage comparable with that normally used in therapy to treat hyperammonemia or to improve metabolisms in patients affected by cirrhosis and stable, overt, chronic hepatic encephalopathy.

  1. Point 4. The number of independent experiments, the number of dishes/wells/cells etc in the performed experiments are not disclosed. As the bar of the controls is without a whisker in most graphs (no variability), it suggests n=1 at least for the controls.

Answer to point 4.:We apologise for omitting all the details related to the number of independent experiments, the number of dishes/wells/cells in the experiments performed. As the bar of controls of figures 1-3 expresses the mean of percentage value, in most of the graphs it is without a whisker since the variability is extremely poor. The omitted information are now indicated in the revised version of the manuscript in the Material and Methods’ section and in the figures of the manuscript. (Page 2; lines 91-97)

  1. Point 5. The statistics are not described at all, rendering the evaluation of the findings also impossible.

Answer to point 5.:We apologise for omitting all the statistical analysis for all the experiments performed. This information is now indicated in the revised version of the manuscript in the Material and Methods’ section. (Page 5; lines 206-215).

Reviewer 4 Report

The current study demonstrates how LOLA can prevent mitochondrial dysfunction using neurodegenerative models. In my opinion, this study is a noteworthy contribution to the field. I have minor points that need to be addressed by the authors: 

  1. The mitochondrial ROS reduction in ROT treated cells by LOLA treatment is very subtle to state (Fig 2 E). This might need to be validated using a better and sensitive method, by flow-cytometry analysis where the sample size is greater than the microscopic methods.  

  1. The western blot quality is very poor in NCX1 detection (Fig 3 C). The reader could hardly identify the band separation in the image. Contrastingly, the other blot using the same antibody (NCX1) gave a better quality blot in Fig 3D.  

Author Response

Referee # 4  

  1. Point 1: The mitochondrial ROS reduction in ROT treated cells by LOLA treatment is very subtle to state (Fig 2 E). This might need to be validated using a better and sensitive method, by flow-cytometry analysis where the sample size is greater than the microscopic methods.

Answer to point 1: We thank the referee for its important suggestion to validate the effect of LOLA on ROS reduction in cells exposed to ROT with a better and sensitive method like flow-cytometry in order to improve the sample size, however, as indicated in the revised version of the manuscript, the confocal microscopy analysis performed by using the fluorescent dye MOTOSOX has been performed on at least 200 cells in three different experimental section. Therefore, the sample size may be considered adequate and statistically significant.

  1. Point 2: The western blot quality is very poor in NCX1 detection (Fig 3 C). The reader could hardly identify the band separation in the image. Contrastingly, the other blot using the same antibody (NCX1) gave a better quality blot in Fig 3D.

Answer to Point 2: We apologize for the quality of the images in Figure 3 that have been modified as requested.

Round 2

Reviewer 2 Report

 The authors address all my concerns. I don't have the other comments.

Author Response

Thank you so much for revising the manuscript “L-ORNITHINE L-ASPARTATE RESTORES MITOCHONDRIAL FUNCTION AND MODULATES INTRACELLULAR CALCIUM HOMEOSTASIS IN PARKINSON’S DISEASE MODELS”. Herein attached you will find the final version of the manuscript containing the required modification of the figure 4, that is now centred and resized, and the required revision of punctuation, spacing and font size that have been carefully corrected, accordingly to your request, throughout the text.

Reviewer 3 Report

no further comments

Author Response

(The authors gave the same response as above.)
